# Fast and Memory Efficient Differentially Private-SGD via JL Projections*

**Zhiqi Bu**
University of Pennsylvania
zbu@sas.upenn.edu

**Sivakanth Gopi**
Microsoft Research
sigopi@microsoft.com

**Janardhan Kulkarni**
Microsoft Research
jakul@microsoft.com

**Yin Tat Lee**
University of Washington
yintat@uw.edu

**Judy Hanwen Shen**
Stanford University
jhshen@stanford.edu

**Uthaipon Tantipongpipat**
Twitter
uthaipon@gmail.com

## Abstract

Differentially Private-SGD (DP-SGD) of [ACG+16] and its variations are the only known algorithms for private training of large scale neural networks. This algorithm requires computation of per-sample gradients norms which is extremely slow and memory intensive in practice. In this paper, we present a new framework to design differentially private optimizers called DP-SGD-JL and DP-Adam-JL. Our approach uses Johnson–Lindenstrauss (JL) projections to quickly approximate the per-sample gradient norms without exactly computing them, thus making the training time and memory requirements of our optimizers closer to that of their non-DP versions. Unlike previous attempts to make DP-SGD faster which work only on a subset of network architectures or use compiler techniques, we propose an algorithmic solution which works for *any* network in a *black-box* manner which is the main contribution of this paper. To illustrate this, on IMDb dataset, we train a Recurrent Neural Network (RNN) to achieve good privacy-vs-accuracy tradeoff, while being significantly faster than DP-SGD and with a similar memory footprint as non-private SGD.

## 1 Introduction

Over the past decade, machine learning algorithms based on (deep) neural architectures have lead to a revolution in applications such as computer vision, speech recognition and natural language processing (NLP). An important factor contributing to this success is the abundance of data. For most of these applications, however, the training data comes from individuals, often containing personal and sensitive information about them. For example, natural language models for applications such as suggested replies for e-mails and dialog systems rely on the training of neural networks on email data of users [CLB+19, DBS19], who may be left vulnerable if personal information is revealed. This could happen, for example, when a model generates a sentence or predicts a word that can potentially reveal private information of users in the training set. Many studies have shown successful membership inference attacks on deep learning models [SSSS17, CLE+19]. Indeed, in a recent work, [CLE+19] show that "unintended memorization" in neural networks is both commonplace and hard to prevent. Such memorization is not due to overtraining [TLL95, CLE+19], and ad hoc techniques such as early-stopping, dropout etc., do not prevent the risk of privacy violations. Moreover, [Fel20] shows that memorization is in fact *necessary*, provably, for some learning tasks. Thus, to prevent unintended privacy breaches one needs a principled approach for private training of deep learning

---

*Author ordering is alphabetical. Work done when the first and the last two authors were interns at Algorithms group, Microsoft Research Redmond.

models. In this paper we study training neural networks with differential privacy, a mathematically rigorous notion of privacy introduced in the seminal work of [DMNS06], and focus on user level privacy.

**Definition 1.1** (($\varepsilon, \delta$)-DP). *We say that an algorithm $M$ is ($\varepsilon, \delta$)-DP if for any two neighboring databases $D, D'$ and any subset $S$ of outputs, we have $\Pr[M(D) \in S] \leq e^{\varepsilon} \Pr[M(D') \in S] + \delta$.*

Besides being a *provable* privacy notion, it has been shown that deep learning models trained with DP protect against leakage of sensitive information; we refer the readers to [CLE+19, ACG+16] for more details.

In a highly influential line of work, [SCS13, BST14, ACG+16] introduced a differentially private version of stochastic gradient descent (DP-SGD) for training deep learning models, and showed that it is possible to achieve reasonable accuracy-vs-privacy tradeoff on common benchmarks such as MNIST and CIFAR10. Since then, there has been a vast body of work building on and extending the DP-SGD algorithm; we refer the readers to [MRTZ18, BDLS19, CLE+19, TAM19, AMR+20, ZWB20, CWH20, BKM+20]. The DP-SGD and its variations such as DP-Adam differ from their non-private counter parts in two crucial ways:

- **Gradient Clipping:** In each iteration of DP-SGD, we clip *each* per-sample gradient to have $\ell_2$-norm at most some fixed parameter $C$. This step ensures that the *sensitivity* of the average gradient is bounded, which is crucial for privacy analysis. Computing the norms of per-sample gradients is the most expensive step in the algorithm. Efficient implementations of backpropagation such as in TensorFlow and PyTorch only maintain the average of per-sample gradients across a batch by default. Therefore, getting the norm of each per-sample gradient requires significantly more time and memory.

- **Adding Noise:** Once clipped gradients are averaged across a batch, DP-SGD algorithm adds carefully calibrated noise, typically sampled from the Gaussian distribution, to ensure privacy.

The analysis of DP-SGD in [ACG+16] then follows from a careful tracking of privacy budget lost in each iteration, for which they introduced a novel technique called *moments accountant*, which was later generalized as Renyi differential privacy by [Mir17]. Tighter and accurate privacy accountants based on the notion of *privacy loss random variables* were given in [KJH20, KJPH21].

While DP-SGD has been shown to achieve reasonable accuracy-vs-privacy tradeoff [BDLS19, ACG+16], and arguably is the only known algorithm for training deep neural networks, its use in real-world deep learning has been rather limited. One of the primary reasons for this is the training time of DP-SGD compared to SGD. In DP-SGD, per-sample gradients are computed at a heavy cost in runtime, especially for large deep learning models. The naive approach of setting the batch size to 1 is too slow to be practical as we completely lose the benefits of parallelization. This problem has attracted significant attention in the community and has been noted in popular implementations of DP-SGD including [TP] and [Opa] (Pytorch DP). Many strategies have been proposed to circumvent this issue, and they fall broadly into the following categories:

- **Microbatching:** DP-SGD implementation in Tensorflow Privacy allows dividing a batch into several microbatches and clipping the gradient at the microbatch level; the per-sample gradients in a microbatch are first averaged and then clipped, and finally these clipped gradients are averaged across microbatches. Thus, if each microbatch is of size $L$, then it gives a speedup of $L$ over the usual DP-SGD. Unfortunately, the sensitivity goes up by a factor of $L$, so we need to add $L$ times more noise. In our experiments, we observe that this often leads to poor accuracy even for moderate values of $L$.

- **Multiple method:** In this approach, proposed by [Goo] and implemented as the vectorized DP-SGD in [TP], one copies the model as many times as there are samples in a batch. As each copy of the model is used on only one example, we can compute the per-sample gradients in parallel. This approach improves speed at the cost of memory and is impractical for large models.

- **Outer product method:** This strategy was proposed by [Goo15] for fully-connected networks and later generalized by [RMT19] to include convolutional layers. The norms of per-sample gradients are computed *exactly* using outer products between the activations and

backpropagated gradients across adjacent layers. In a very recent work, [LK21] showed how to extend this approach to recurrent layers. A drawback of this approach is that it does not work for all network architectures in a black-box manner, and needs careful implementation for each network architecture. Furthermore, the current implementations of these methods require significantly more memory than non-private SGD as shown in [SVK20].

- **Compiler Optimization:** A completely different approach towards mitigating the problem was suggested by [SVK20]. They showed that by exploiting language primitives, such as vectorization, just-in-time compilation, and static graph optimization, one can implement DP-SGD algorithm significantly faster. They demonstrated these ideas in two frameworks: JAX and TensorFlow. While we believe this is exciting progress, the ideas in [SVK20] are specific to these JAX and TensorFlow implementations (as of today) and present a non-algorithmic approach to this problem.

Table 1: Summary of different methods for DP training of neural networks

| Optimizers | Privacy | Speed | Memory | Generalizability |
|---|---|---|---|---|
| Non-DP SGD | ✗ | ✔ | ✔ | ✔ |
| DP-SGD-Vanilla | ✔ | ✗ | ✔ | ✔ |
| DP-SGD-Multiple | ✔ | ✔ | ✗ | ✔ |
| DP-SGD-Outer | ✔ | ✔ | ✗ | ✗ |
| JAX | ✔ | ✔ | ✔ | ✗ |
| DP-SGD-JL | ✔ | ✔ | ✔ | ✔ |

As summarized in Table 1, none of these approaches for speeding up DP-SGD completely solve the problem and fall short in at least one dimension. In this work, we propose a new *algorithmic framework* based on JL-projections for *fast and memory efficient* differentially private training of deep neural networks, which bypasses the expensive step of exactly computing per-sample gradient norms.

## 1.1 Our Contributions and Techniques

The main idea of our algorithm is to *approximate* the per-sample gradient norms instead of computing them exactly. Johnson–Lindenstrauss (JL) projections provide a convenient way to approximate the $\ell_2$ norm of a vector; simply project the vector onto a uniformly random direction, the length of the projection (scaled appropriately) is a good approximation to the $\ell_2$-norm of the vector. By doing more such projections and averaging, we get even better approximation to the true $\ell_2$-norm. Moreover, there is an efficient way to obtain such projections using forward-mode auto-differentiation or Jacobian-vector product (`jvp`) (see Section 2.1 for details). `jvp` can be calculated during the forward pass making it very efficient. Since this makes the *sensitivity* itself a random variable, the privacy analysis is significantly harder than the traditional DP-SGD. Moreover the Moments Accountant and related Renyi DP Accountant are not applicable to analyze the privacy of our algorithm since the associated privacy loss random variables have unbounded moments. Instead, we convolve the privacy loss distributions using Fast Fourier Transform (FFT) as done in [KJH20].

To summarize, the key contributions of this paper are:

- Our algorithms DP-SGD-JL and DP-Adam-JL are considerably faster than previously known differentially private training algorithms that require exact per sample gradient norms, and work for *all* network architectures. The privacy-vs-accuracy tradeoff achieved by our algorithms is comparable to the existing state-of-the-art DP-algorithms.

- Memory footprint of our algorithms is nearly the same as that of non-private training algorithms. This allows us to work with larger batch sizes (which is crucial for achieving good privacy-vs-accuracy tradeoffs) without resorting to gradient accumulation. This also improves the running time of training. This is because we never need to calculate (and store) all the per-sample gradients simultaneously, whereas DP-SGD will need to store all the per-sample gradients simultaneously.

- Compared to DP-SGD, our analysis of privacy is more involved. Since we only approximate the per-sample gradient norms, we cannot precisely bound *sensitivity*. Therefore the analysis requires significantly new ideas, which could be of independent interest.

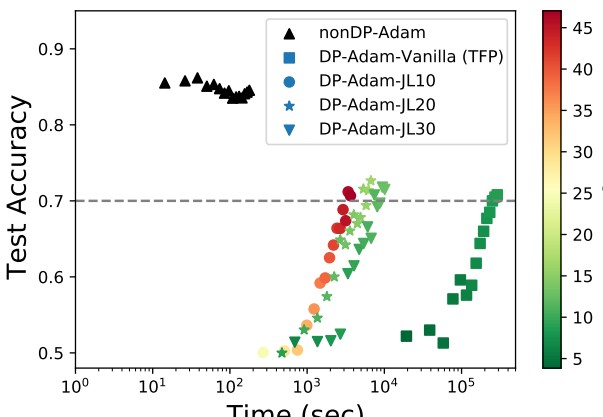

Figure 1: Performance of various algorithms training an RNN on IMDb dataset using a batch size of 256. The privacy parameter $\varepsilon$ is color coded for $\delta = 10^{-5}$. In DP-Adam-JL$r$, $r$ refers to the JL dimension, i.e., the number of JL projections used to approximate per-sample gradient norms. DP-Adam-Vanilla (TFP) is the implementation of DP-Adam in [TP] library.

- We demonstrate these improvements by training an RNN using layers such as bidirectional LSTM, embedding layer, fully connected etc. on the IMDb dataset. As can be seen from Figure 1, our algorithms are significantly faster than current implementations of DP-SGD while achieving similar privacy-vs-accuracy tradeoff.

- Our algorithms introduce a new *knob*, the dimension of JL-projection, which allows us to do a tradeoff between training time and privacy, which was not possible in earlier algorithms. All hyperparameters being the same, smaller JL dimension will give much better running time with an increase in privacy budget.

- Our experiments show that although the privacy bounds we could prove for DP-SGD-JL are not so great for very small JL dimensions (see Figure 3), their behavior (accuracy-vs-epoch curve) converges very quickly to that of DP-SGD-Vanilla. Figure 4 shows that DP-SGD-JL(3) is already very close to DP-SGD-Vanilla and DP-SGD-JL(20) is almost indistinguishable. We find these properties of DP-SGD-JL algorithms to be very useful during initial stages of experimentation and hyper-parameter tuning for private training.

## 2 DP-SGD-JL Algorithm

In this section we describe our new differentially private optimizers. We will first present DP-SGD-JL, and DP-Adam-JL follows in the same lines and is presented in the Supplementary Material. We begin with an introduction to "Jacobian-vector product" (jvp) which is crucial for our algorithm.

### 2.1 Jacobian-vector product (jvp)

Given a function $f : \mathbb{R}^d \to \mathbb{R}$, the gradient of $f$ with respect to $\theta$, denoted by $\nabla_\theta f$, is:

$$\nabla_\theta f = \left( \frac{\partial f}{\partial \theta_1}, \frac{\partial f}{\partial \theta_2}, \dots, \frac{\partial f}{\partial \theta_d} \right).$$

Let $F : \mathbb{R}^d \to \mathbb{R}^m$ be some function given by $F(\theta) = (F_1(\theta), F_2(\theta), \dots, F_m(\theta))$. The Jacobian of $F(\theta)$, denoted by $\nabla_\theta F$, is the matrix:

$$\nabla_\theta F = \left[ \frac{\partial F_i}{\partial \theta_j} \right]_{ij} = \begin{bmatrix} \nabla_\theta F_1 \\ \nabla_\theta F_2 \\ \vdots \\ \nabla_\theta F_m \end{bmatrix}.$$

Most auto-differentiation packages allow for calculating the vector-Jacobian product (vjp) given by $u^T \nabla_\theta F = \sum_{i=1}^m u_i \nabla_\theta F_i$ for any $u \in \mathbb{R}^m$ efficiently using reverse-mode auto-differentiation, which

is the familiar 'backpropagation'.[2] One can also calculate the Jacobian-vector product (`jvp`) given by

$$\nabla_\theta F \cdot v = \begin{bmatrix} \langle \nabla_\theta F_1, v \rangle \\ \langle \nabla_\theta F_2, v \rangle \\ \vdots \\ \langle \nabla_\theta F_m, v \rangle \end{bmatrix}$$

efficiently using forward-mode auto-differentiation (i.e., the required derivatives are calculated during the forward pass of the network). We refer the reader to the survey on automatic differentiation by [BPRS17] for more details. `jvp` is implemented using forward-mode auto-differentiation in the recent TensorFlow versions.[3] Unfortunately, PyTorch doesn't have an implementation of forward-mode auto-differentiation. Instead, one can compute `jvp` using two calls to `vjp`, this is called the 'double `vjp` trick' (see [Tow]).[4] Define $G(\alpha) = \alpha^T \nabla_\theta F$, which can be calculated using `vjp`. Note that $\nabla_\alpha G = (\nabla_\theta F)^T$. Now we can use `vjp` again on $G$ to calculate `jvp` as

$$v^T \nabla_\alpha G = v^T (\nabla_\theta F)^T = (\nabla_\theta F \cdot v)^T.$$

In our experiments, we use the efficient implementation of `jvp` in TensorFlow to compute the Jacobian-vector products as the double `vjp` trick is a few times slower.

## 2.2 Algorithm

The main idea of our algorithm is to approximate $\ell_2$-norms of per-sample gradient norms instead of computing them exactly. And the key tool to achieve this is JL projections.

**Proposition 2.1** (JL projections). *Let $y \in \mathbb{R}^d$ be any vector. If $v_1, v_2, \ldots, v_r \sim \mathcal{N}(0, I_d)$ are independent standard Gaussian vectors, then $M_r = \sqrt{\sum_{i=1}^{r} \frac{1}{r} \langle y, v_i \rangle^2}$ has the same distribution as $\|y\|_2 \sqrt{\frac{1}{r} \chi_r^2}$.[5] In particular $\mathbb{E}[M_r^2] = \|y\|_2^2$.*

*Proof.* By the properties of the standard Gaussian distribution, $\langle y, v_i \rangle$ has the distribution of $\|y\|_2 \mathcal{N}(0, 1)$. And $\langle y, v_i \rangle$ are independent for $i = 1$ to $r$. Therefore $\sum_{i=1}^{r} \langle y, v_i \rangle^2$ has the same distribution as $\|y\|_2^2 \chi_r^2$. ∎

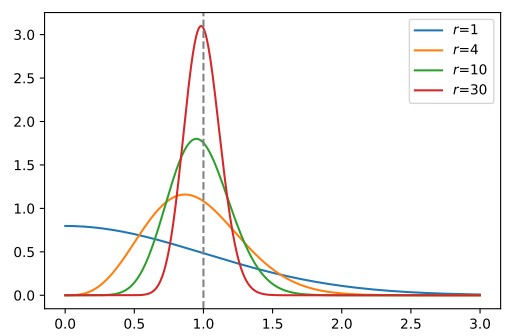

Figure 2: Distribution of $\sqrt{\frac{1}{r} \chi_r^2}$ for different $r$.

As shown in Figure 2, as $r$ grows larger, the distribution of $\sqrt{\frac{1}{r} \chi_r^2}$ concentrates more around 1 and therefore $M_r$ becomes a better estimate of $\|y\|_2$. Using `jvp`, we can compute projections of per-sample gradients on to standard Gaussian vectors quickly and therefore get good approximations to their norms. This is the main idea of DP-SGD-JL (Algorithm 1).

## 2.3 Privacy Analysis

Let us first recall the privacy analysis of DP-SGD in Abadi *et al.* [ACG+16], which consists of three main arguments: 1) First, they show that each step of the DP-SGD algorithm is differentially private,

---

[2]In PyTorch, $u^T \nabla_\theta F$ can be calculated as `autograd.grad(F,θ,grad_outputs=u)`. In TensorFlow, this is `tf.GradientTape().gradient(F,θ,output_gradients=u)`.

[3]Supported in tf-nightly≥2.4.0.dev20200924 as `tf.autodiff.ForwardAccumulator(θ,v).jvp(F)`. JAX also has an implementation of `jvp`.

[4]In Pytorch, an implementation of `jvp` using the double `vjp` trick exists and can be invoked as `torch.autograd.functional.jvp(F,θ,inputs=v)`.

[5]$\chi_r^2$ is the chi-square distribution with $r$ degrees of freedom which is the distribution of sum of squares of $r$ standard Gaussians.

**Algorithm 1:** Differentially private SGD using JL projections (DP-SGD-JL)

---

**Input:** Examples $\{x_1, x_2, \ldots, x_N\}$, loss function $\mathcal{L}(\theta) = \mathbb{E}_{i \in [N]}[\mathcal{L}(\theta; x_i)]$, initialization $\theta_0$.
Parameters: number of iterations $T$, learning rates $(\eta_1, \eta_2, \ldots, \eta_T)$, noise scale $\sigma$, batch size $B$,
clipping norms $(C_1, C_2, \ldots, C_T)$, number of JL projections $r$.

**for** $t = 1$ *to* $T$ **do**

    Sample $S_t = \{X_1, X_2, \ldots, X_B\} \subset \{x_1, x_2, \ldots, x_N\}$ uniformly at random.;

    Define $F(\theta) = (\mathcal{L}(\theta; X_1), \mathcal{L}(\theta; X_2), \ldots, \mathcal{L}(\theta; X_B))$;

    Sample $v_1, v_2, \ldots, v_r \leftarrow \mathcal{N}(0, I_d)$ (where $\theta \in \mathbb{R}^d$);

    `// JL projections to estimate per-sample gradient norm`

    **for** $j = 1$ *to* $r$ **do**

        $(P_{1j}, P_{2j}, \ldots, P_{Bj}) \leftarrow \nabla_\theta F(\theta_{t-1}) \cdot v_j$ ;             `// Computed using jvp`

        `// Note that` $P_{ij} = \langle \nabla_\theta \mathcal{L}(\theta_{t-1}; X_i), v_j \rangle$

    **for** $i = 1$ *to* $B$ **do**

        Set $M_i = \sqrt{\frac{1}{r} \sum_{j=1}^{r} P_{ij}^2}$ ;         `//` $M_i$ `is an estimate for` $\|\nabla_\theta \mathcal{L}(\theta_{t-1}; X_i)\|_2$.

    `// Scale the losses to implicitly clip per-sample gradients`

    Define $\widetilde{\mathcal{L}}(\theta) = \frac{1}{B} \sum_{i \in B} \min\{1, \frac{C_t}{M_i}\} \cdot \mathcal{L}(\theta; X_i)$;

    `//` $\nabla_\theta \widetilde{\mathcal{L}}(\theta_{t-1})$ `is the average of clipped gradients, computed by one backprop`

    `// Add noise to` $\nabla_\theta \widetilde{\mathcal{L}}(\theta_{t-1})$`, the average of clipped gradients`

    $\tilde{g}_t \leftarrow \nabla_\theta \widetilde{\mathcal{L}}(\theta_{t-1}) + \frac{\sigma \cdot C_t}{B} \cdot \mathcal{N}(0, I_d)$;

    Update $\theta_t \leftarrow \theta_{t-1} - \eta_t \tilde{g}_t$;

**Output:** $\theta_0, \theta_1, \theta_2, \ldots, \theta_T$

---

which follows from the simple observation that norm of the each gradient can be at most $C$ due to clipping and so each step is a Gaussian mechanism. 2) Next, they observe that each mini-batch is a random sample of the dataset, hence one can appeal to the privacy amplification by subsampling. 3) Finally, they do a composition across all the iterations of the DP-SGD algorithm. However, existing works at that time did not give a tight analysis of the privacy loss in steps 2 and 3, hence [ACG+16] invented the Moments Accountant (MA) technique.

Unfortunately, analysis of our algorithm is significantly more involved than that of [ACG+16], as all the above 3 steps pose new technical challenges. First note that unlike DP-SGD, in our algorithm, we only have a rough estimate of the norm of each gradient; hence, the sensitivity of the gradients is a random variable. This means that the each iteration is not a Gaussian mechanism. The Moments Accountant (and the closely related RDP Accountant) cannot be used compute the privacy loss of the composition because the privacy loss random variables for our algorithms do not have exponential moments (for JL dimension $r$, only moments $< r/2$ exist).

To overcome these challenges, we use the notion of *privacy loss random variables* (PRVs) [DR16, MM18, KJH20, GLW21]. In particular, we use the PRV Accountant from [GLW21]. Define $\delta(X||Y)(\varepsilon) = \Pr[Y > \varepsilon] - e^\varepsilon \Pr[X > \varepsilon]$. For any DP algorithm $M$, there exists random variables $(X, Y)$, called its privacy loss random variables (PRVs) such that the privacy curve of $M$ is given by $\delta(X||Y)$, i.e., $M$ satisfies $(\varepsilon, \delta(X||Y)(\varepsilon))$-DP for every $\varepsilon$.

The following theorem shows a lower bound on the privacy curve of each iteration of the DP-SGD-JL algorithm, which is the main technical contribution of this paper.

**Theorem 2.2.** *Let* $Z_r = \frac{1}{\sqrt{\frac{1}{r}\chi_r^2}}$ *where* $\chi_r^2$ *is the chi-square distribution with $r$ degrees of freedom. Each iteration of DP-SGD-JL (before subsampling) satisfies* $(\varepsilon, \delta(X||Y)(\varepsilon))$-*DP where*

$$X = \mathcal{N}\left(-\frac{Z_r^2}{2}, Z_r^2\right) \text{ and } Y = \mathcal{N}\left(\frac{Z_r^2}{2}, Z_r^2\right). \tag{1}$$

From the above theorem, we can prove that the privacy curve of each iteration behaves like $\delta(\varepsilon) \sim \frac{1}{\varepsilon^{r/2}}$ for large $\varepsilon$. The proof of Theorem 2.2 can be found in the supplementary material.

If $M_1, M_2, \ldots, M_k$ are DP algorithms such that $M_i$ has PRVs $(X_i, Y_i)$, then the composition of $M_1, M_2, \ldots, M_k$ has PRVs given by $(\sum_{i=1}^{k} X_i, \sum_{i=1}^{k} Y_i)$ [GLW21]. If a mechanism $M$ has PRVs

given by $(X, Y)$, then the subsampled mechanism $M \circ \mathrm{Sample}_p$ where $\mathrm{Sample}_p$ samples $p$ fraction of the data uniformly at random has PRVs given by $(X_p, Y_p)$ where:

$$
\begin{aligned}
X_p &= \log(1 + p(e^X - 1)), \\
Y_p &= \begin{cases} \log(1 + p(e^Y - 1)) \text{ w.p. } p \\ \log(1 + p(e^X - 1)) \text{ w.p. } 1 - p. \end{cases}
\end{aligned}
\tag{2}
$$

See [GLW21] for a proof of this fact. Combining these facts about PRVs with Theorem 2.2, we can compute the privacy curve of the DP-SGD-JL algorithm.

**Theorem 2.3.** *Let $p = B/N$ where $B$ is the batch size and $N$ is the total number of samples. Then DP-SGD-JL run for $T$ iterations satisfies $(\varepsilon, \delta(\varepsilon)$-DP where*

$$
\delta \equiv \delta \left( \sum_{i=1}^{T} X_p^i \middle\| \sum_{i=1}^{T} Y_p^i \right)
$$

*where $X_p^1, X_p^2, \ldots, X_p^T$ are iid copies of $X_p$ and $Y_p^1, Y_p^2, \ldots, Y_p^T$ are iid copies of $Y_p$. $X_p$ and $Y_p$ are given by (2) and $X, Y$ are given by (1).*

Finally, we numerically compute the distribution of $\sum_{i=1}^{T} X_p^i$ by using Fast Fourier Transform (FFT) to convolve the distributions of $X_p$ with itself $T$ times. The distribution of $\sum_{i=1}^{T} Y_p^i$ is similarly calculated, see [GLW21] on how to approximate these distributions.

**Effect of JL dimension on privacy**    Figure 3 shows how the JL dimension $r$ effects the privacy curve of DP-SGD-JL. As JL dimension increases, DP-SGD-JL becomes more and more private and converges to the privacy of DP-SGD.

# 3    Experiments

In this section, we demonstrate experimentally that compared to existing implementations of DP-SGD with exact per-sample gradient clipping, our optimizers have significant advantages in speed and memory cost while achieving comparable accuracy-vs-privacy tradeoff. Moreover our algorithms perform well on a variety of network architectures. The main goal of this section is to give empirical evidences towards the following three strengths of our algorithm alluded in the introduction:

1. Our algorithm is significantly faster compared to per-sample gradient computations and works for any network in a black-box way.
2. Memory footprint of our algorithm is roughly same as non-private SGD.
3. The DP-SGD-JL algorithms with smaller values of JL dimension exhibit similar behavior as DP-SGD but with orders of magnitude speed up, and hence can be used for hyper-parameter search.

In the following, we write 'nonDP-SGD' for the standard non-private SGD and 'DP-SGD-Vanilla' for the implementation of DP-SGD in [TP], nonDP-Adam and DP-Adam-Vanilla are similarly defined. We use Tensorflow and [TP] for all our experiments because [Opa] does not support arbitrary network architectures.[6] Moreover Tensorflow has an efficient implementation of jvp while PyTorch doesn't.

**Notation:**    We denote the noise multiplier as $\sigma$, clipping norm as $C$, batch size as $B$, learning rate as $\eta$ and the number of epochs as $E = BT/N$. We fix the privacy parameter $\delta = 10^{-5}$, as done by prior work. We denote the JL dimension used by each optimizer in the parentheses. We use one Tesla P100 16GB GPU for all experiments. In all the experiments, we report the time per epoch by averaging over a large number of epochs. The code for our experiments is available in the supplementary material.

## 3.1    Training an LSTM model on IMDb dataset

The goal of these experiments is to demonstrate the first strength of our algorithm: it is significantly faster than per-sample gradient computations and works for any network in a black-box way. We

---

[6]In Pytorch Opacus github, the LSTM layer is only partially supported, e.g. single directional, single LSTM layer, no dropout layer; other recurrent layers such as GRU are not supported (see Opacus).

| Non-DP | DP-Vanilla | DP-JL(1) | DP-JL(5) | DP-JL(10) | DP-JL(30) |
|--------|------------|----------|----------|-----------|-----------|
| 12     | 19317      | 41       | 128      | 243       | 675       |

Table 2: Seconds per epoch to train our RNN with 598,274 parameters and 25,000 training samples. We set $\beta_1 = 0.9, \beta_2 = 0.999, \sigma = 0.6, C = 1, B = 256, \eta = 0.001, E = 15$. We use Adam as the optimizer.

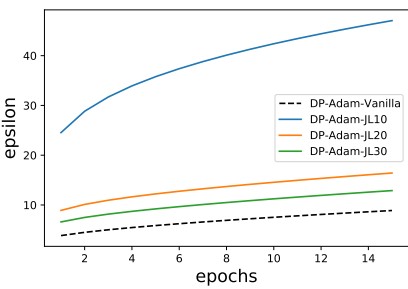

Figure 3: Privacy curves for various algorithms while training the LSTM model on IMDb dataset. Here $\delta = 10^{-5}$ is fixed.

train a bidirectional LSTM with an embedding layer on the IMDb dataset for sentiment analysis. We remark that simpler networks *not* based on RNNs can achieve good accuracy-vs-privacy tradeoff as shown in [BDLS19] and [Pyt]. However, LSTM models based on RNN architectures are widely used in NLP applications, and hence efficient private training of such models remains an important challenge in this area [MRTZ18]. Moreover, as we noted in the introduction, extensions of outer product trick to bidirectional LSTMs as described in [LK21] are significantly more complicated, and require considerable effort to implement. Since authors of [LK21] did not provide the code, we are unable to compare the improvements. Moreover, as also noted in [LK21], the outer product method requires significantly more memory and hence will not scale to large batch sizes, which is very important to achieve good privacy vs utility tradeoff.

We implement DP-Adam-JL using `jvp` method in TensorFlow. We train the same single-layer bidirectional LSTM[7] as in the [Ten] tutorial, using the same IMDb dataset with 8k vocabulary. The dataset has binary labels and is preprocessed by zero-padding the sequences to a length of 150 before being fed into the embedding layer.

Table 2 shows the training time per epoch for different algorithms. As expected, we observe that DP-Adam-JL, referred to simply as DP-JL in the table, algorithms with smaller values of JL dimension are significantly faster than DP-Adam-Vanilla. We also notice a linear increase in running-time with incresing JL dimension. However, as we note in the Figure 3 privacy guarantees of DP-Adam-JL algorithm with smaller values of JL dimension are considerably worse than DP-Adam-Vanilla. On the other hand, DP-Adam-JL(30) is $30\times$ faster than DP-Adam-Vanilla while achieving similar privacy guarantees as DP-Adam-Vanilla. When allowed to train for sufficient number of epochs, we observed that all the algorithms achieved same accuracy but with different privacy costs and running times. This three dimensional tradeoff between utility, privacy and speed is depicted in the Figure 1 (see introduction), where we plot the privacy values using a color plot.

**Remark 3.1.** *As we can see from Table 2, for a batch size of 256, the slowdown of DP-Adam-Vanilla is more than 256 compared to nonDP-Adam. This is counter intuitive as a naive implementation DP-Adam-Vanilla, by setting the batch size equal to 1 and then doing gradient accumulation across 256 batches should only be 256 times slower. As we show below, this is due to the memory issues of DP-Adam-Vanilla in the implementation in [TP]. Indeed the naive implementation of DP-Adam-Vanilla in tensorflow using gradient accumulation takes about 4000 seconds per epoch for the same batch size.*

## 3.2 Memory footprint

Another key strength of JL based algorithms is their memory consumption, and the goal of this section is to show this aspect of our new algorithms via experiments. As a proxy for memory consumption,

---

[7]We cannot use CuDNNLSTM and instead use LSTM for the following reason. When using CuDNNLSTM (and on GPU), we observe a significant speedup compared to LSTM, but the accuracy is invalid and we further incur a LookupError when computing `jvp`.

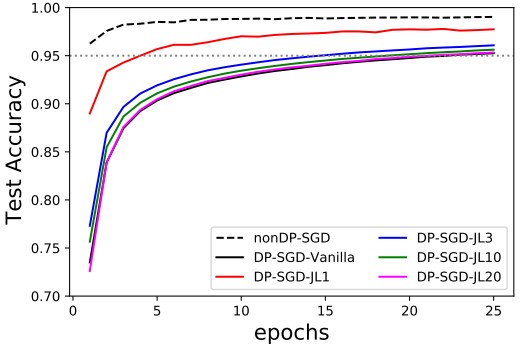

Figure 4: Behavior of various private algorithms for MNIST dataset. The test accuracy is averaged over 25 runs. Note that the DP-SGD-JL20 curve is nearly overlapping with the curve for DP-SGD-Vanilla.

we compare the largest batch size each algorithm can handle without running out of memory. It is known that to achieve good privacy-vs-accuracy tradeoffs for DP-SGD, we need to use large batch sizes [ACG+16]. One way to support large batch sizes is via gradient accumulation; however, this has the disadvantage that one loses parallelism, which in turns leads to slower run times. Hence memory footprint of algorithms also indirectly affects the training time.

We compare our JL algorithms with the implementation of DP-SGD-Vanilla in [TP]. We train a convolutional neural network from [TP] tutorial on MNIST dataset, which has 60,000 training samples.[8] As we can see from Table 3 (left), DP-SGD-JL algorithm and nonDP-SGD can both run with the maximum possible batch size of 60,000 whereas DP-SGD-Vanilla can only handle a batch size of at most 500. In general, we believe that the memory footprint of DP-SGD-JL algorithm is very close to that of non-private SGD. To show this, we augment the CNN in [TP] tutorial with dense layers to blowup the model size to 17,146,938 parameters, and repeat the experiment. As we see from Table 3 (right), the largest batch size supported by DPSGD-JL(30) is only a factor 2 away from the largest batch size supported by non-DP SGD. On the other hand, we observe that DP-SGD-Vanilla only supports a batch size of 100.

| nonDP-SGD | 60,000 | nonDP-SGD | 52,000 |
|---|---|---|---|
| DP-SGD-Vanilla | 512 | DP-SGD-Vanilla | 100 |
| DP-SGD-JL(10) | 60,000 | DP-SGD-JL(10) | 28,000 |
| DP-SGD-JL(30) | 60,000 | DP-SGD-JL(30) | 25,000 |

Table 3: The left table shows the maximum batch size supported by various algorithms on a CNN with 26,010 parameters trained on MNIST with 60,000 training samples. The right table shows the maximum batch size supported by various algorithms on a CNN with 17,146,938 parameters trained on MNIST with 60,000 training samples. Training done on one Tesla P100 16GB GPU.

The above experiments also give a possible explanation of why DP-SGD-Vanilla implementation in TFP has a slowdown that is larger than the batch size, as we observed in the LSTM experiments. Even for MNIST, we observe that DP-SGD-Vanilla running time gets better with batch size in the very beginning but as the batch size becomes larger the running time gets worse, and soon after it runs out of memory.

### 3.3 Using DP-SGD-JL for hyper-parameter search

As we saw in our experiments summarized in Table 2, our algorithms with small values of JL dimension are orders of magnitude faster than DP-Adam-Vanilla; DP-Adam-JL(1) is about 470x faster and DP-SGD-JL(5) is about 150x faster. However, unfortunately, the privacy bounds we can prove for these algorithms are considerably worse than DP-Adam-Vanilla. Despite this drawback, we observe that behavior of DP-SGD-JL even for small JL dimension is very close to that of DP-Adam-Vanilla. Figure 4 plots the accuracy vs epochs for various algorithms training a CNN from [TP] tutorial on MNIST dataset. We observe that as JL dimension increases, the accuracy vs epoch

---

[8]We use the implementation and the network from `mnist_dpsgd_tutorial_keras.py`

curve quickly converges to that of DP-SGD-Vanilla. DP-SGD-JL(3) is already very similar to DP-SGD-Vanilla and DP-SGD-JL(20) is nearly indistinguishable. This also lets us hypothesize that the privacy of our algorithms could be much better than what we could prove and that it should converge equally quickly to that of DP-SGD-Vanilla. Thus we believe that DP-SGD-JL(3) or DP-SGD-JL(5) are good candidates for experimentation and hyper-parameter tuning during private training, since their behavior is almost identical to that of DP-SGD-Vanilla while being orders of magnitude faster.

## Acknowledgments and Disclosure of Funding

We thank Sergey Yekhanin for his constant support and encouragement during this work. We also thank Lukas Wutschitz and Osman Ramadan for helpful discussions. Finally, we thank the amazing open source community of TensorFlow for their quick response in fixing bugs which was crucial for our experiments ([TFi]).

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
