# A  DP-SGD and DP-Adam-JL

For completeness, we provide pseudo-code for DP-SGD and DP-Adam-JL used in our experiments. DP-Adam-JL satisfies exactly the same privacy bounds as DP-SGD-JL.

---

**Algorithm 2:** Differentially private Adam using JL projections (DP-Adam-JL)

---

**Input:** Examples $\{x_1, x_2, \ldots, x_N\}$, loss function $\mathcal{L}(\theta) = \mathbb{E}_{i \in [N]}[\mathcal{L}(\theta; x_i)]$, initialization $\theta_0$. Parameters: number of iterations $T$, momentum parameters $(\beta_1, \beta_2)$, learning rate $\eta_t$, a small constant $\xi > 0$, noise scale $\sigma$, batch size $B$, clipping norms $(C_1, C_2, \ldots, C_T)$, number of JL projections $r$. All operations on vectors are element-wise.

Define $m_0 = 0$ and $u_0 = 0$;
**for** $t = 1$ *to* $T$ **do**
    Sample $S_t = \{X_1, X_2, \ldots, X_B\} \subset \{x_1, x_2, \ldots, x_N\}$ uniformly at random;
    Define $F(\theta) = (\mathcal{L}(\theta; X_1), \mathcal{L}(\theta; X_2), \ldots, \mathcal{L}(\theta; X_B))$;
    Sample $v_1, v_2, \ldots, v_r \leftarrow \mathcal{N}(0, I_d)$ (where $\theta \in \mathbb{R}^d$);
    // JL projections to estimate per-sample gradient norm
    **for** $j = 1$ *to* $r$ **do**
        $(P_{1j}, P_{2j}, \ldots, P_{Bj}) \leftarrow \nabla_\theta F(\theta_{t-1}) \cdot v_j$ ;        // Computed using jvp
        // Note that $P_{ij} = \langle \nabla_\theta \mathcal{L}(\theta_{t-1}; X_i), v_j \rangle$
        ;
    **for** $i \in B_t$ **do**
        $M_i \leftarrow \sqrt{\frac{1}{r} \sum_{j=1}^{r} P_{ij}^2}$ ;        // $M_i$ is an estimate for $\|\nabla_\theta \mathcal{L}(\theta_{t-1}; X_i)\|_2$.
    // Scale the losses implicitly to clip per-sample gradients
    Define $\widetilde{\mathcal{L}}(\theta) = \frac{1}{B} \sum_{i \in B} \min\{1, \frac{C_t}{M_i}\} \cdot \mathcal{L}(\theta; X_i)$ ;
    // $\nabla_\theta \widetilde{\mathcal{L}}(\theta_{t-1})$ is the average of clipped gradients, computed by one backprop
    // Add noise to $\nabla_\theta \widetilde{\mathcal{L}}(\theta_{t-1})$, the average of clipped gradients
    $\tilde{g}_t \leftarrow \nabla_\theta \widetilde{\mathcal{L}}(\theta_{t-1}) + \frac{\sigma \cdot C_t}{B} \cdot \mathcal{N}(0, I_d)$;
    $m_t \leftarrow \beta_1 m_{t-1} + (1 - \beta_1) \tilde{g}_t$;
    $u_t \leftarrow \beta_2 u_{t-1} + (1 - \beta_2) \tilde{g}_t^2$;
    $\hat{m}_t \leftarrow m_t / (1 - \beta_1^t)$;
    $\hat{u}_t \leftarrow u_t / (1 - \beta_1^t)$;
    Update $\theta_t \leftarrow \theta_{t-1} - \eta_t \hat{m}_t / (\sqrt{\hat{u}_t} + \xi)$;
**Output:** $\theta_0, \theta_1, \theta_2, \ldots, \theta_T$

---

**Algorithm 3:** Differentially private SGD (DP-SGD) from [ACG$^+$16]

---

**Input:** Examples $\{x_1, x_2, \ldots, x_N\}$, loss function $\mathcal{L}(\theta) = \mathbb{E}_{i \in [N]}[\mathcal{L}(\theta; x_i)]$, initialization $\theta_0$. Parameters: number of iterations $T$, learning rates $(\eta_1, \eta_2, \ldots, \eta_T)$, noise scale $\sigma$, batch size $B$, clipping norms $(C_1, C_2, \ldots, C_T)$.
**for** $t = 1$ *to* $T$ **do**
    Sample $S_t = \{X_1, X_2, \ldots, X_B\} \subset \{x_1, x_2, \ldots, x_N\}$ uniformly at random;
    **for** $i = 1$ *to* $B$ **do**
        $g_i \leftarrow \nabla_\theta \mathcal{L}(\theta_{t-1}; X_i)$;
        // Clip the per-sample gradients to $\ell_2$-norm at most $C_t$
        $\hat{g}_i \leftarrow \min\{1, \frac{C_t}{\|g_i\|_2}\} \cdot g_i$;
    // Average the clipped gradients and add noise
    $\tilde{g}_t \leftarrow \frac{1}{B} \left( \sum_{i \in B_t} \hat{g}_i \right) + \frac{\sigma C_t}{B} \cdot \mathcal{N}(0, I_d)$;
    Update $\theta_t \leftarrow \theta_{t-1} - \eta_t \tilde{g}_t$;
**Output:** $\theta_0, \theta_1, \ldots, \theta_T$

---

## B   Privacy Analysis

**Definition B.1** (Privacy curve [GLW21]). *Given two random variables $X, Y$ supported on some set $\Omega$, define $\delta(X||Y) : \mathbb{R} \to [0, 1]$ as:*

$$\delta(X||Y)(\varepsilon) = \sup_{S \subset \Omega} \Pr[Y \in S] - e^{\varepsilon} \Pr[X \in S].$$

**Proposition B.2** (Post-processing [DRS19, GLW21]). *Let $X, Y$ be two random variables supported on $A$ and let $M : A \to B$ is some randomized function, then $\delta(X||Y) \geq \delta(M(X)||M(Y))$.*

**Proposition B.3.** *Let $(X_1, Y_1)$ and $(X_2, Y_2)$ be pairs of random variables such that $X_1|_{Y_1=y}$ has the same distribution as $X_2|_{Y_2=y}$ for all $y$. Then $\delta(X_1, Y_1||X_2, Y_2) = \delta(Y_1||Y_2)$.*

*Proof.* By post-processing (Proposition B.2), $\delta(X_1, Y_1||X_2, Y_2) \geq \delta(Y_1||Y_2)$. Let $X(y)$ be a random variable which has the distribution of $X_1|_{Y_1=y}$ and $X_2|_{Y_2=y}$. Let $M(y) = (X(y), y)$. Then $M(Y_1) = (X_1, Y_1)$ and $M(Y_2) = (X_2, Y_2)$. Therefore, by post-processing (Proposition B.2), we have the inequality in the other direction. $\qquad\square$

**Proposition B.4** ([GLW21]). *For any privacy curve $\delta(P||Q)$ between two random variables $P, Q$, there exist random variables $X, Y$ called Privacy Loss Random Variables (PRVs) such that:*

1. *$\delta(P||Q) = \delta(X||Y)$*

2. *$\delta(X||Y) = \Pr[X \geq \varepsilon] - e^{\varepsilon} \Pr[Y \geq \varepsilon] = \mathbb{E}_Y[(1 - e^{\varepsilon - Y})_+]$.*

*Moreover $X, Y$ are given by:*

$$X = \log\left(\frac{Q(\omega)}{P(\omega)}\right) \text{ where } \omega \sim P,$$

$$Y = \log\left(\frac{Q(\omega)}{P(\omega)}\right) \text{ where } \omega \sim Q$$

*where $P(\cdot), Q(\cdot)$ are probability density functions of $P, Q$ respectively.*

The following lemma is very useful for our privacy analysis.

**Lemma B.5.** *Let $Z$ be any random variable. Then the PRVs for the privacy curve $\delta(Z, N(Z, 1)||Z, N(0, 1))$ are given by:*

$$X = \mathcal{N}\left(-\frac{Z^2}{2}, Z^2\right) \text{ and } Y = \mathcal{N}\left(\frac{Z^2}{2}, Z^2\right). \tag{3}$$

*Proof.* Denote $P := Z \times N(Z, 1), Q := Z \times N(0, 1)$. By Proposition B.4,

$$
\begin{aligned}
Y &= \log\left(\frac{Q(z, x)}{P(z, x)}\right) \text{ where } (z, x) \sim Q \\
&= \log\left(\frac{Z(z) \exp(-x^2/2)}{Z(z) \exp(-(x-z)^2/2)}\right) \text{ where } z \sim Z, x \sim N(0, 1) \\
&= \frac{(x-z)^2}{2} - \frac{x^2}{2} \text{ where } z \sim Z, x \sim N(0, 1) \\
&= -xz + \frac{z^2}{2} \text{ where } z \sim Z, x \sim N(0, 1) \\
&= N\left(\frac{Z^2}{2}, Z^2\right).
\end{aligned}
$$

A similar calculation gives $X$. $\qquad\square$

We will now prove an other lemma which is useful for our privacy analysis.

**Lemma B.6.** *Let $Z, \widetilde{Z}$ be two random variables such that there exists some coupling $(Z, \widetilde{Z})$ with $Z \geq \widetilde{Z} \geq 0$. Then $\delta(Z, N(Z, 1)||Z, N(0, 1)) \geq \delta(\widetilde{Z}, N(\widetilde{Z}, 1)||\widetilde{Z}, N(0, 1))$.*

*Proof.* We will prove this by post-processing (Proposition B.2). Define a randomized map

$$M(z, a) = \left( \widetilde{z}, \left( \frac{\widetilde{z}}{z} \right) a + N\left( 0, 1 - \left( \frac{\widetilde{z}}{z} \right)^2 \right) \right)$$

where $\widetilde{z} \sim \widetilde{Z}|_{Z=z}$, i.e., $\widetilde{z}$ is sampled from the conditional distribution of $\widetilde{Z}$ given $Z = z$. Note that $0 \leq \widetilde{z} \leq z$ because the coupling $(Z, \widetilde{Z})$ satisfies $0 \leq \widetilde{Z} \leq Z$. Now it is easy to verify that $M(Z, N(Z, 1)) = (\widetilde{Z}, N(\widetilde{Z}, 1))$ and $M(Z, N(0, 1)) = (\widetilde{Z}, N(0, 1))$. $\square$

## B.1 Proof of privacy for Algorithm 1

---
**Algorithm 4:** Subroutine of Algorithm 1

---
**Input:** Vectors $\{g_1, g_2, \ldots, g_N\} \subset \mathbb{R}^d$, clipping norm $C$, noise scale $\sigma$, number of JL projections $r$

```
// JL projections to estimate per-sample gradient norm
```
Sample $v_1, v_2, \ldots, v_r \leftarrow \mathcal{N}(0, I_d)$;

For $i \in [N]$, set $M_i = \sqrt{\frac{1}{r} \sum_{j=1}^{r} \langle g_i, v_j \rangle^2}$ ;          `// `$M_i$` is an estimate for `$\|g_i\|_2$

```
// Average clipped per-sample gradients and add noise
```
$\tilde{g} \leftarrow \left( \sum_{i=1}^{N} \min\{1, \frac{C}{M_i}\} \cdot g_i \right) + \sigma \cdot C \cdot \mathcal{N}(0, I_d)$;

**Output:** $\tilde{g}$

---

We will first analyze the privacy of a crucial subroutine used in Algorithm 1 which is shown in Algorithm 4.

**Theorem B.7.** *Algorithm 4 satisfies $(\varepsilon, \delta(\varepsilon))$-DP where*

$$\delta \equiv \delta(Z_r/\sigma, \mathcal{N}(Z_r/\sigma, 1) \| Z_r/\sigma, \mathcal{N}(0, 1))$$

*and $Z_r = \frac{1}{\sqrt{\frac{1}{r}\chi_r^2}}$. The PRVs for the privacy curve $\delta(\varepsilon)$ are given by:*

$$X = \mathcal{N}\left( -\frac{Z^2}{2\sigma^2}, \frac{Z^2}{\sigma^2} \right) \ and \ Y = \mathcal{N}\left( \frac{Z^2}{2\sigma^2}, \frac{Z^2}{\sigma^2} \right). \tag{4}$$

*Proof.* Let $X$ be the output of Algorithm 4 with input $\{g_0, g_1, \ldots, g_N\}$ and let $Y$ be the output of Algorithm 4 with input $\{g_1, \ldots, g_N\}$. We want to show that $\delta(X\|Y) \leq \delta$. We have

$$X = \left( \sum_{i=0}^{N} \min\left\{1, \frac{C}{M_i}\right\} \cdot g_i \right) + \sigma \cdot C \cdot \mathcal{N}(0, I_d), \ \ Y = \left( \sum_{i=1}^{N} \min\left\{1, \frac{C}{M_i}\right\} \cdot g_i \right) + \sigma \cdot C \cdot \mathcal{N}(0, I_d).$$

By post-processing property (Proposition B.2), we have

$$\delta(X\|Y) \leq \delta(v_1, v_2, \ldots, v_r, M_0, X' \| v_1, v_2, \ldots, v_r, M_0, Y')$$

where

$$X' = \min\left\{1, \frac{C}{M_0}\right\} \cdot g_0 + \sigma \cdot C \cdot \mathcal{N}(0, I_d), \ \ Y' = \sigma \cdot C \cdot \mathcal{N}(0, I_d).$$

By Proposition B.3,

$$\delta(v_1, v_2, \ldots, v_r, M_0, X' \| v_1, v_2, \ldots, v_r, M_0, Y') = \delta(M_0, X' \| M_0, Y').$$

Let $U$ be a rotation matrix which rotates $g_0$ to $\|g_0\|_2 \cdot e_1 \in \mathbb{R}^d$ where $e_1 = (1, 0, 0, \ldots, 0)$. Let $X'' = UX'$ and $Y'' = UY'$. Since $U$ is a fixed bijective map,

$$\delta(M_0, X' \| M_0, Y') = \delta(M_0, X'' \| M_0, Y'').$$

Because of rotation invariance, $U \cdot \mathcal{N}(0, I_d)$ has the same distribution as $\mathcal{N}(0, I_d)$. So,

$$X'' = \min\left\{1, \frac{C}{M_0}\right\} \cdot \|g_0\|_2 \, e_1 + \sigma \cdot C \cdot \mathcal{N}(0, I_d), \ \ Y'' = \sigma \cdot C \cdot \mathcal{N}(0, I_d).$$

The coordinates $(X_i'')_{i \geq 2}$ are independent of each other and $M_0, X_1''$. Similarly the coordinates $(Y_i'')_{i \geq 2}$ are also independent of each other and $M_0, Y_1''$. Moreover $X_i''$ and $Y_i''$ has the same distribution for $i \geq 2$. Therefore,

$$\delta(M_0, X'' || M_0, Y'') = \delta(M_0, X_1'' || M_0, Y_1''),$$

where $X_1'' = \min\left\{1, \frac{C}{M_0}\right\} \cdot \|g_0\|_2 + \sigma \cdot C \cdot \mathcal{N}(0,1)$ and $Y_1'' = \sigma \cdot C \cdot \mathcal{N}(0,1)$.

Let $\phi(M_0) = \min\left\{\frac{\|g_0\|_2}{\sigma C}, \frac{\|g_0\|_2}{\sigma M_0}\right\}$. We can further simplify this using Proposition B.3 as:

$$\begin{aligned}
&\delta(M_0, X_1'' || M_0, Y_1'') \\
&= \delta\left(M_0, \phi(M_0) + \mathcal{N}(0,1) \,||\, M_0, \mathcal{N}(0,1)\right) \\
&= \delta\left(M_0, \phi(M_0), \phi(M_0) + \mathcal{N}(0,1) \,||\, M_0, \phi(M_0), \mathcal{N}(0,1)\right) \\
&= \delta\left(\phi(M_0), \phi(M_0) + \mathcal{N}(0,1) \,||\, \phi(M_0), \mathcal{N}(0,1)\right)
\end{aligned}$$

We can simplify $\phi(M_0)$ further by using the fact that $\frac{\|g_0\|}{M_0} = \|g_0\|_2 / \left(\sqrt{\frac{1}{r}\sum_{j=1}^r \langle g_0, v_j\rangle^2}\right)$ has the same distribution as $1/\sqrt{\frac{1}{r}\chi_r^2}$. Therefore $\phi(M_0)$ has the same distribution as

$$\tilde{Z} = \min\left\{\frac{\|g_0\|_2}{\sigma C}, \frac{1}{\sigma\sqrt{\frac{1}{r}\chi_r^2}}\right\}.$$

Let $Z_r = \frac{1}{\sqrt{\frac{1}{r}\chi_r^2}}$ and so $\tilde{Z} = \min\left\{\frac{\|g_0\|_2}{\sigma C}, \frac{Z_r}{\sigma}\right\}$. By Lemma B.6,

$$\delta(\tilde{Z}, \mathcal{N}(\tilde{Z}, 1) || \tilde{Z}, \mathcal{N}(0,1)) \leq \delta(Z_r/\sigma, \mathcal{N}(Z_r/\sigma, 1) || Z_r/\sigma, \mathcal{N}(0,1)).$$

Therefore, this proves that $\delta(X||Y) \leq \delta(Z_r/\sigma, \mathcal{N}(Z_r/\sigma, 1) || Z_r/\sigma, \mathcal{N}(0,1))$ where $Z_r = \frac{1}{\sqrt{\frac{1}{r}\chi_r^2}}$. The PRVs follow from Lemma B.5. $\qquad\square$

## B.2  Effect of JL dimension on privacy

Since the per-sample gradient norm estimations get more accurate with JL dimension, it is clear that the privacy of DP-SGD-JL should converge to that of DP-SGD-Vanilla for large JL dimension. We also observe that privacy parameter $\varepsilon$ is monotonically decreasing with increasing JL dimension and eventually converges to the $\varepsilon$ for DP-SGD-Vanilla. This can be see from Figure 3.