# OpenReview forum: "Fast and Memory Efficient Differentially Private-SGD via JL Projections"
_NeurIPS.cc/2021/Conference — NeurIPS 2021 Poster_

### Official Review · Reviewer_occM · 2021-07-16

**Rating:** 6
**Confidence:** 3

**Summary:**

To accelerate DP-SGD, this paper uses JL projection to approximate the $\ell_2$ norm of the per-sample gradients, which can be calculated during the forward pass efficiently. Moreover, the privacy analysis requires new techniques (convolving the privacy loss distributions using FFT) since the standard moment accountant cannot be used in this case.

**Limitations And Societal Impact:**

See above.

**Main Review:**

- The proposed method does not explicitly compute nor store the per-sample gradients, unlike DP-SGD.
- The novelty and technical depth of this paper is limited.
- The experiments show that the accuracy of the proposed method may be worse than DP-SGD (see Figure 3). although much faster.

**Time Spent Reviewing:**

1

---

> ### Author Response · Authors · 2021-08-10
> **Thanks for the feedback**
>
> Thanks for your positive review.   We are glad that you found our privacy analysis using FFT interesting. We appreciate it.

---

### Official Review · Reviewer_GLcg · 2021-07-17

**Rating:** 6
**Confidence:** 3

**Summary:**

This paper considers the computation and memory cost problem introduced by per-sample clipping in private gradient descent algorithms. To accelerate per-sample clipping, this paper proposed a new framework using JL projections to estimate the per-sample gradient norms. The authors provide rigorous differential privacy analysis of the proposed framework. They also give empirical evaluations of the proposed algorithms in terms of privacy, utility, and computation time.

**Limitations And Societal Impact:**

Yes

**Main Review:**

Strengths:

- This idea of using JL projections to estimate the gradient norms and using Jacobian-vector product to accelerate computation is new. The framework is potentially useful for training large-scale neural networks.
- This paper is technically sound. The authors discussed that each iteration of the proposed algorithm is not a Gaussian mechanism due to the estimated clipping norm. Thus, they provided rigorous privacy analysis for the new framework.
- This paper is well written and clearly presented.

Weaknesses and some discussion:
- The main concern is that the privacy bound provided in this paper of this framework is larger than that of the standard differentially private SGD algorithms as shown in Figure 1. I am wondering if the privacy bound or the numeric method to compute privacy can be improved?

- In Figure 3, If we further increase the projection dimension r to match the privacy level of DP-Adam, how does the computation time compare with that of DP-Adam/DP-SGD? Will DP-Adam-JL still require less computation time than DP-Adam?


Overall, I think this paper is above borderline. The significance can be improved by improving the privacy bound of the proposed framework, so I recommend borderline acceptance.

**Time Spent Reviewing:**

8

---

> ### Author Response · Authors · 2021-08-10
> **Thanks for your feedback and effort in reviewing the paper.**
>
> Thanks for the positive review and helpful comments and suggestions. We appreciate your effort and time spent in reviewing this paper. Now we address some of your comments mentioned in weaknesses.
> 1.	Tighter privacy analysis. We agree with you that it might be possible to improve the privacy analysis in the paper. But the numerical method to compute the privacy bounds proved in the paper is nearly tight. The improvement must come by better analysis of per iteration privacy. We believe that this is an exciting open problem. If one can improve per iteration privacy analysis, then we may be able to match the privacy bounds of vanilla DPSGD using much lower JL dimensions such as 10. This has potential to lead to orders of magnitude savings in speed. But it is worth noting that despite this, our compute cost savings using JL dimension of 20 or 30 is still substantial from a practical perspective as shown in Figure 1.
> 2.	Matching privacy of DPSGD.  Thanks for your question. The run time vs JL dimension tradeoff for our algorithms is nearly linear. So JL60 should take about 1400secs per epoch while the privacy nearly matches that of vanilla-DPSGD. But the runtime of vanilla-DPSGD is ~19000 secs per epoch. We will include this experiment and explanation in our next version.

---

### Official Review · Reviewer_SDe9 · 2021-07-29

**Rating:** 3
**Confidence:** 4

**Summary:**

This paper presents a new algorithm to approximate per-sample gradient norms in DP-SGD using Johnson-Lindenstrauss (JL) projection. Experimental results were conducted on IMDb and MNIST datasets showing that the proposed approach is faster with the same memory footprint as the vanilla DP-SGD.



**Ethical Concerns:**

No concern

**Limitations And Societal Impact:**

The paper did not discuss limitations and societal impact.

**Main Review:**

Approximating the per-sample gradient norm is new and interesting. The paper presentation is clear and easy to understand. However, there is room for improvement.
Figure 3 shows that the proposed approach consumes an excessive privacy budget epsilon compared with the vanilla DP-SGD. In the low-value regime of privacy budget (tight privacy protection), it is unclear how the proposed approach can advance the vanilla DP-SGD. This is quite critical in practice since the trade-off between model utility and privacy loss in the low-value regime of the privacy budget is more significant. From that angle, the contribution of the paper is marginal.
An extensive study is needed to show the advantages of the proposed approach clearly. Larger datasets are needed since it is hard to draw a convincing conclusion only using the MNIST dataset.
Question: Whether all the algorithms in Figure 4 consume the same privacy budget given the same number of epochs? It appears that they do not consume the same privacy budget given the analysis in Figure 3. As a result, the comparison in Figure 4 is unconvincing.

**Time Spent Reviewing:**

4

---

> ### Author Response · Authors · 2021-08-10
> **Thanks for the feedback**
>
> Thanks for your review and feedback. We appreciate your time and effort. We now address the main comments.
>
> 1.	Your summary of our contributions: We respectfully disagree with the summary provided. Our experiments show that our algorithm DPSGD-JL is significantly faster than vanilla-DPSGD. But its memory footprint is comparable to “non-private” SGD, whereas vanilla-DPSGD consumes significantly more memory than non-private SGD. Based on our experience in training large models, we feel that this is an important consideration that cannot be ignored.
>
> 2.	Comments regarding Figure 4. There seems to be some misunderstanding regarding Figure 4. The figure is not comparing privacy-vs-utility tradeoffs for DPSGDJL algorithm with DPSGD Vanilla. Figure 4 is intended to demonstrate that our JL algorithms can be used for hyperparameter search for vanilla DPSGD. Hyperparameter tuning is an expensive operation for large models and is itself an active area of research. It is the most time and resource consuming step for training large models. Figure 4 shows that DPSGDJL algorithm with small dimension behaves almost identical to vanilla DPSGD in terms of training dynamics when instantiated with the same hyperparameters. But DPSGDJL with JL dimension = 3 is significantly faster than DPSGD vanilla as we demonstrate in our experiments. This suggests that we can use DPSGDJL for fast hyperparameter tuning for private training, while saving both time and compute cost.
>
> For the privacy-vs-utility tradeoffs and privacy curve comparisons, please see Figure 1 and Figure 3.
>
> 3.	Accuracy-vs-Utility Tradeoffs: We agree that when JL dimension is small (10 or less), our privacy bounds are worse than DPSGD.  However, by increasing the JL dimension, we can achieve comparable privacy-vs-utility tradeoffs compared to DPSGD. As you can see from Figure 1 in the introduction, for JL dimension 20 and 30, we are very close to the privacy-vs-utility tradeoff of DPSGD, while being considerably faster and using substantially less memory.
>
> 4.	High privacy regime or epsilons used in real world applications: Almost all the current deployments of privately trained deep learning models we are aware of use $\epsilon \approx 3$ to $\epsilon \approx 7$ (single digit epsilons), since the utility is too bad in the high privacy regime, i.e., for $\epsilon < 1$. Furthermore, all the previous works on deep learning with differential privacy, starting from the seminal work of Abadi et al, also used epsilons in this range. So, we focus on the single digit epsilon regime of privacy.

---

### Official Review · Reviewer_3iUF · 2021-07-31

**Rating:** 7
**Confidence:** 4

**Summary:**

Differentially private stochastic gradient descent requires clipping the gradient computed on each example before aggregation. That is, we compute the 2-norm of the gradient vector and, if it is too large, we scale it down. Surprisingly, this step entails a significant computational overhead relative to non-private stochastic gradient descent.
The reasons for this slowdown are not entirely clear to me. It seems to be a software issue deep in the bowels of TensorFlow and PyTorch -- somewhere between the Python and the Hardware this operation is poorly optimized.

This paper proposes to alleviate the problem by performing approximate, rather than exact clipping. Specifically, a Johnson-Lindenstrauss sketch is used to approximate the 2-norm of the vector instead of exactly computing it. This JL approximation can be computed more efficiently than the exact norm. However, there is a tradeoff between speed and approximation quality as we can alter the dimension of the JL sketch.

Technically, the most involved aspect of this work is the privacy analysis. Since the gradients are only approximately bounded, we cannot apply the usual analysis. This algorithm does not satisfy Renyi differential privacy as there is heavy-tailed privacy loss due to the possibility of drastically underestimating the norm. Instead an entirely new analysis is required based on the recent Fourier accounting approach.

Experimental results demonstrate that there is a significant improvement in runtime. However, the privacy parameters also degrade significantly.


**Limitations And Societal Impact:**

Yes.

**Main Review:**

This paper presents a novel approach to accelerating differentially private deep learning. This is accompanied by a thorough theoretical privacy analysis and an experimental evaluation.

Computation is a significant bottleneck in practical implementations of differentially private deep learning. Thus the faster approach in this paper is valuable. However, it comes at the expense of a blowup in the privacy parameters. This tradeoff may be acceptable in some circumstances, but unacceptable in others. In any case, it is a good addition to our toolbox.

One thing to note is that the problem being solved is, as far as I know, a software issue, rather than a fundamental limitation. Thus the value of the proposed solution may disappear with a future version of the software.


A low level comment: Differentially private stochastic gradient descent predates the work of Abadi et al. I believe it was first introduced by Bassily, Smith, & Thakurta in 2014. Abadi et al. however were the first to do it in a practical setting.

**Time Spent Reviewing:**

3

---

> ### Author Response · Authors · 2021-08-10
> **Thanks for the positive feedback.**
>
> Thanks for the positive review and feedback.  We are glad that you find our privacy analysis based on Fourier approach interesting. Thanks also for pointing out the paper of Bassily et al., we will cite this paper in the next revision.  One small comment regarding the problem studied. While we agree that deep learning libraries in the future can indeed provide APIs for calculating per-sample gradients, if one must store all the per-sample gradients explicitly then the memory cost will be huge. Further, we also feel that our algorithm with small JL dimension of 3 or 4 can be informative for hyper-parameter search with very little computational cost compared to running DPSGD.
>
> Thanks again for your review and we appreciate it.

---

### Decision · Program_Chairs · 2021-09-27

**Decision:**

Accept (Poster)

**Comment:**

This submission makes significant time and memory efficiency improvements for differentially private SGD using random projections, at the cost of a worse privacy/utility tradeoff when compared with vanilla DPSGD as emphasized by one reviewer. The authors gave a compelling rebuttal however that despite this worse tradeoff, DPSGDJL still has value for fast hyperparameter tuning for private training. The contribution of this submission thus seems strong.